# Long-term temporal evolution of extreme temperature in a warming Earth

**Justus Contzen** [1,2] *, **Thorsten Dickhaus** [3], **Gerrit Lohmann** [1,2]

**1** Section Paleoclimate Dynamics, Alfred Wegener Institute Helmholtz Centre for Polar and Marine Research, Bremerhaven, Germany, **2** Department of Environmental Physics, University of Bremen, Bremen, Germany, **3** Institute for Statistics, University of Bremen, Bremen, Germany

* justus.contzen@awi.de

**Data Availability Statement:** The Data used in this study are publicly available in the Copernicus Climate Data Store (DOI: 10.24381/cds.d3513dbf). The code that was used in this analysis is publicly

## Abstract

We present a new approach to modeling the future development of extreme temperatures globally and on the time-scale of several centuries by using non-stationary generalized extreme value distributions in combination with logistic functions. The statistical models we propose are applied to annual maxima of daily temperature data from fully coupled climate models spanning the years 1850 through 2300. They enable us to investigate how extremes will change depending on the geographic location not only in terms of the magnitude, but also in terms of the timing of the changes. We find that in general, changes in extremes are stronger and more rapid over land masses than over oceans. In addition, our statistical models allow for changes in the different parameters of the fitted generalized extreme value distributions (a location, a scale and a shape parameter) to take place independently and at varying time periods. Different statistical models are presented and the Bayesian Information Criterion is used for model selection. It turns out that in most regions, changes in mean and variance take place simultaneously while the shape parameter of the distribution is predicted to stay constant. In the Arctic region, however, a different picture emerges: There, climate variability is predicted to increase rather quickly in the second half of the twenty-first century, probably due to the melting of ice, whereas changes in the mean values take longer and come into effect later.

## Introduction

In many regions of the world, a rising trend in frequency and magnitude of temperature extremes is currently observed ([1–3]). Heatwaves and extreme temperatures can have devastating effects on human societies and ecosystems ([4–6]) as well as on economies and agriculture ([7, 8]). The consequences of an increase in frequency or magnitude of extreme events can also be considerably more severe than those of changes in mean temperature alone ([9]), explaining why the investigation of climate extremes is an increasingly active research topic ([10–14]). However, the development of temperature extremes in the future decades and centuries on a global level is still less well understood, and the focus of most studies is on regional investigations or on the near future ([15]). It has been observed that changes in extreme

available on Zenodo (DOI: 10.5281/zenodo. 7413932).

**Funding:** JC is funded through the Helmholtz School for Marine Data Science (https://www. mardata.de; grant no. HIDSS-0005). GL receives funding through "Ocean and Cryosphere under climate change" in the Program "Changing Earth – Sustaining our Future" of the Helmholtz Society (https://www.helmholtz.de/en/about-us/structure-and-governance/program-oriented-funding/) and through PalMod by the Bundesministerium für Bildung und Forschung (https://www.palmod.de/; grant no. 01LP1917A). The funders had no role in study design, data collection and analysis, decision to publish, or preparation of the manuscript.

**Competing interests:** The authors have declared that no competing interests exist.

temperature are not taking place uniformly around the globe, but that they are instead showing a strong dependency on the geographic location and its climatic conditions. This is visible already on a regional level ([16], [17]) and even more globally ([18]). Earth system model simulations predict this variability also to be present in the future development of Earth's climate ([19], [20]).

Changes in the expected frequency of extreme events can be caused by changes in various statistical parameters, like the mean and the variance ([21]). In addition to that, starting time and duration of changes can also vary in different regions. In our work, we create statistical models to investigate changes in temperature extremes in a warming climate on a global scale and for a period of investigation spanning several centuries. In order to get insights into the questions outlined above, four Earth system models will be analyzed with respect to daily temperatures from historical and future simulations ranging from 1850 to 2300.

It is expected that the rate of change of extremes will increase in the near future ([22]). Under the premise that mankind will be able to slow and ultimately end the increase of atmospheric $CO_2$ emissions someday, it can be expected that in consequence, changes in extreme temperatures will gradually slow down as the climate system will be tending toward a new equilibrium state ([23]), although it may still take centuries for a new stationary state to be completely reached due to slow-changing components of the climate system ([24]). Taking these considerations together, we can expect changes in extreme temperature to follow in general a slow—fast—slow pattern over time. To describe a transition from an initial value to a final one that starts slowly, then speeds up and finally decelerates again when approaching the new value, it is common practice to use a logistic function, which exhibits a characteristic S-shaped form. The first application of logistic functions in modeling is due to Verhulst, who designed a logistic growth model to describe the development of biological populations in 1845 ([25]). The motivation in the ecological context is that the population growth is slow at the beginning (limited by the small population size) as well as at the end (limited by the lack of natural resources). The logistic growth model has been successfully applied in biology and epidemiology—a recent example being its application to the spreading of the coronavirus desease 2019 ([26])—and this has motivated its use as a general model to describe changes from one state to another in fields as varied as linguistics ([27]), medicine ([28]) or economics ([29]).

The analysis of extremes is complicated by the fact that extreme events are often rare, and that it is therefore difficult to build informative statistics based solely on the extreme events themselves. One common approach to overcome this issue is based on block-wise maxima: The data are split up into different (time) blocks of a sufficiently large size and then the maxima of each block are investigated. Under suitable conditions, the distribution of the block-wise maxima can be approximated by a generalized extreme value (GEV) distribution ([30], [31]). GEV distributions have found numerous applications in climatology and hydrology, examples include [32–34]. A GEV distribution is determined by three parameters, called "location", "scale" and "shape", with the latter one describing the heavy-tailedness of the distribution. To model extremes in a changing climate, we will use non-stationary GEV distributions with time-dependent distribution parameters. The changes in the distribution parameters will be described using logistic functions. After fitting the statistical models to the data, we will analyze the estimated distribution parameters in detail, and we will use the estimates also to investigate futue changes in the distribution quantiles.

Changes in the expected frequency of extreme events can be caused by changes in the mean values of the GEV distributions, changes in their variability, changes in their heavy-tailedness or by a combination of these factors ([35–37]). The application of non-stationary GEV distributions enables us to investigate which factors contribute to what extent at different

geographic locations. In addition, we will investigate whether changes in the different distribution parameters occur simultaneously or if changes in some parameters precede changes in others.

Several non-stationary models based on GEV distributions have been proposed to describe the influence of climate change on climate extremes: In [38], a GEV distribution with the parameters polynomially depending on time was proposed and its application was showcased using precipitation data from Greece. In a similar way, in [39], non-stationary models with different degrees of freedom were constructed and evaluated using Bayesian inference and Markov chain Monte Carlo techniques. In [40], an idea first proposed in [41] was extended and neural networks were used to choose between a variety of non-stationary models with different covariates that can interact with each other. The approach of combining GEV distributions with logistic functions gives us the possibility to investigate developments in extreme temperature over a time span of several centuries and on a global level and to research how changes in extreme temperatures will unfold in different regions.

The rest of this paper is organized as follows: In the next section, the temperature data sets and the logistic models as well as the model-fitting algorithm are presented. The results of applying the logistic models to the data are shown in the section thereafter. In addition to that, results of a simulation study that is conducted to investigate the accuracy of the model fitting algorithm are also discussed there. The section is followed by a discussion section, and a section on conclusions and an outlook finalize the article.

## Data and methods

### Data

We investigate daily temperature data at two meters above surface from four global earth system models. For each earth system model, the data consist of a simulation of the historical climate from 1850 to 2005 and a future simulation from 2005 to 2300 that follows the representative concentration pathway RCP8.5 of the Intergovernmental Panel on Climate Change IPCC ([42]). The RCP8.5 scenario provides atmospheric $CO_2$ values until the year 2100. For the years after 2100, the climate model runs with prescribed $CO_2$ values that are set to the value of the year 2100, see Fig 1a. The four Earth system models used are the model bcc-csm1–1 from the Beijing Climate Center (in the following: "BCC"; [43]), the model CCSM4 from the National Center for Atmospheric Research NCAR ("CCSM4", [44]), the CSIRO-Mk3–6-0 ("CSIRO", [45]), and the MPI-ESM-LR from the Max Planck Institute for Meteorology in Hamburg, Germany ("MPI-ESM", [46]). All models take part in the Climate Model Intercomparison Project CMIP5 ([47]). In the plots, the coastline boundaries shown have been obtained from Natural Earth Version 4.2.0 (http://www.naturalearthdata.com/).

In Fig 1b, the evolutions of the annual global mean temperature that are predicted by the four Earth system models are displayed. They roughly follow an S-shaped form for each model, but differ strongly among the different climate models in terms of timing and magnitude of the changes.

### The block maxima approach

The statistical models we develop and apply in this work are based on the well-established block-maxima approach, for which we will now briefly present the theoretical foundation. Let $X_1, \ldots, X_n$ be stochastically independent random variables, each having the same (unknown) probability distribution. We investigate the distribution of the maximum of the variables: $Y^{(n)} := \max_{i = 1, \ldots, n}(X_i)$. We assume the existence of suitable normalizing sequences $(a_n)_{n \in \mathbb{N}}$

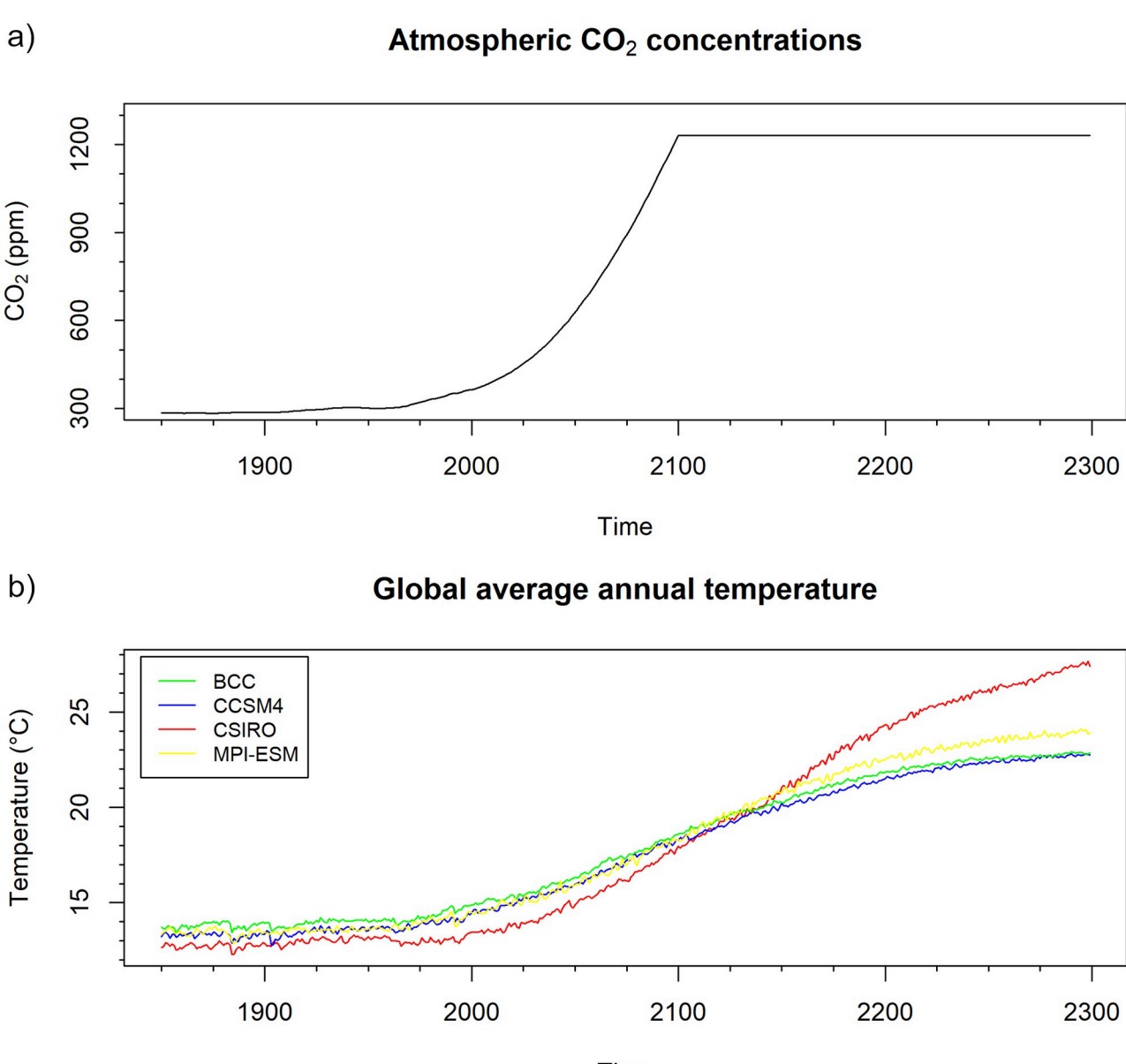

**Fig 1. Atmospheric CO$_2$ concentration and global annual mean temperature.** Panel a: The atmospheric CO$_2$ concentration (in ppm) that was used for the model runs. The CO$_2$ concentration follows the RCP8.5 scenario ([42]) until 2100 and is kept constant afterwards. Panel b: The annual global mean temperature (in ˚C) according to the climate model runs.

and $(b_n)_{n\in\mathbb{N}}$ with $a_n > 0$ for all $n$ such that these block-wise maxima converge in distribution as the block size $n$ tends to infinity:

$$\frac{Y^{(n)} - b_n}{a_n} \xrightarrow{\mathcal{D}} H. \tag{1}$$

It is shown in [30, 31, 48] that in this case, $H$ must follow a GEV distribution. The GEV distribution has three parameters: location ($\mu$), scale ($\sigma > 0$) and shape ($\gamma$), and its cumulative

distribution function is given by

$$
F_{\mu,\sigma,\gamma}(x) = \begin{cases} \exp\left(-\exp\left(-\dfrac{x-\mu}{\sigma}\right)\right) & \gamma = 0 \\[2ex] \exp\left(-\max\left(0, 1 + \gamma\dfrac{x-\mu}{\sigma}\right)^{-\frac{1}{\gamma}}\right) & \gamma \neq 0. \end{cases} \tag{2}
$$

While location and scale parameters correspond very roughly to mean and standard deviation, the shape parameter is a measure of the heavy-tailedness of the distribution. Location and scale parameter of the distribution of $H$ depend on the choice of $(a_n)_{n\in\mathbb{N}}$ and $(b_n)_{n\in\mathbb{N}}$, while the shape parameter does not. It is therefore justified to say that $X$ is in the domain of attraction of a unique shape parameter value $\gamma$ if Eq (1) is fulfilled for some normalizing sequences $(a_n)_{n\in\mathbb{N}}$, $(b_n)_{n\in\mathbb{N}}$ and some random variable $H$ following a GEV distribution with shape parameter $\gamma$.

In our application, $n$ is fixed as the number of days in a year. Motivated by Eq (1), we calculate for each year annual maxima $y_1^{(n)}, y_2^{(n)}, \ldots$, and we approximate the distribution of these maxima by a GEV distribution (where the constants $a_n$ and $b_n$ are absorbed into the parameters $\{\lambda, \sigma, \gamma\}$).

A GEV distribution with a shape parameter $\gamma$ greater than 0 is also called a Fréchet distribution and is heavy-tailed (i.e. it features strong positive extremes that are markedly different from the non-extreme values). A GEV distribution with $\gamma = 0$ is called a Gumbel distribution and has exponential tails. The GEV distributions with $\gamma < 0$ (Weibull distributions) have a finite right endpoint. For a more in-depth introduction to GEV distributions and the block-maxima approach, see [49], Chapter 7.

The family of GEV distributions has been widely used in climatology as a model for block-wise maximized data, often applied to the yearly maxima of daily average temperature ([50–52]), an approach which we follow as well. Note that we do not need to do an adjustment for seasonality, because in the presence of a strong seasonality, the yearly maxima are selected from the season with the warmest temperature anyway.

In climate data, one extreme event can occur over a time span that includes a boundary between two blocks. The extreme event can then be responsible for two dependent consecutive block maxima. An approach to overcome this issue is to introduce a fixed time span $\tau$ that is assumed to be a typical duration of an extreme event and then to adapt the maxima selection process in order to ensure that the selected values from the blocks are at least a time span of $\tau$ apart. Specifically, if for two consecutive blocks the block maxima are less than $\tau$ apart, the lower one of these maxima is discarded and is replaced by the block maximum based on only those values that are distant enough from the maximum of the other block. This method, which was developed first in [53] and then employed for example in [54, 55], will also be applied to our data, using a time span $\tau$ of 30 days.

## Models for non-stationary GEV parameters

To model the effects of changes in the climate, the GEV distributions we use need to have time-dependent distribution parameters. Due to the reasoning laid out in the introduction, we choose logistic functions to describe the change of the GEV parameters over time. The logistic function we use as the basis for our models is given by

$$
f(x) = \frac{1}{1 + \exp(-2 \cdot \log(19) \cdot x)}. \tag{3}
$$

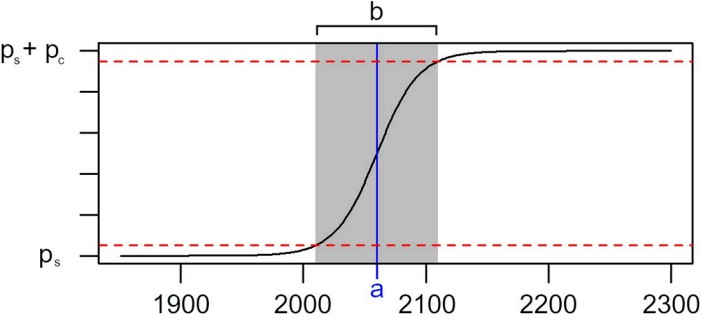

**Fig 2. Visualization of the parameter values of the logistic models.** A sigmoidal curve following Eq (4) with parameters $a$ = 2060 and $b$ = 100 is displayed. Parameter $a$ corresponds to the time point at which half of the transition from $p_s$ to $p_s + p_c$ is completed. Ninety percent of this transition take place within the interval $\left[a - \frac{b}{2}, a + \frac{b}{2}\right]$, so parameter $b$ describes the approximate time span of the transition.

It describes a growth limited by 0 for $x \to -\infty$ and by 1 for $x \to \infty$ with the highest growth rate at $x$ = 0. The constant 2·log(19) in the exponential function is used for better interpretability of the parameters of the models we will present below, it ensures that 90% of the change from 0 to 1 takes place in the interval $\left[-\frac{1}{2}, \frac{1}{2}\right]$. We use the function in our models in the following way: For each of the three GEV parameters $p \in \{\mu, \sigma, \gamma\}$ we describe its temporal development as

$$\hat{p}(t) = p_s + p_c \cdot f\left(\frac{t - a}{b}\right). \tag{4}$$

The model parameter $p_s$ describes the "initial state" and $p_c$ describes the total magnitude of the change. The model parameters $a$ and $b$ control the timing of the change. Parameter $a$ indicates the time point at which the growth rate is highest (which is also the time point at which exactly half of the change from $p_s$ to $p_s + p_c$ is completed) and parameter $b$ indicates the approximate duration of the change (in the sense that 90% of the total change takes place in the time span $\left[a - \frac{b}{2}, a + \frac{b}{2}\right]$). See also Fig 2 for a visualization.

This leads to the following model:

**Model 1a.** The three GEV parameters location, scale and shape are described using a logistic curve, using for each parameter a different initial value and amount of change. The parameters $a$ and $b$ are the same for location, scale and shape.

$$\hat{\mu}(t) = \mu_s + \mu_c \cdot f\left(\frac{t - a}{b}\right)$$

$$\hat{\sigma}(t) = \sigma_s + \sigma_c \cdot f\left(\frac{t - a}{b}\right)$$

$$\hat{\gamma}(t) = \gamma_s + \gamma_c \cdot f\left(\frac{t - a}{b}\right)$$

As pointed out in [56], the evolution of extreme events may be different from that of mean and variance (which may show different behaviors among themselves). It may therefore be necessary to allow for changes in location, scale and shape to take place at different times and over different durations. This leads to the following more complex model:

**Model 1b.** This model is the same as Model 1a, but with individual parameters $a_\mu$, $b_\mu$, $a_\sigma$, $b_\sigma$ and $a_\gamma$, $b_\gamma$ being used for location, shape and scale of the GEV distribution.

$$\hat{\mu}(t) = \mu_s + \mu_c \cdot f\left(\frac{t - a_\mu}{b_\mu}\right)$$

$$\hat{\sigma}(t) = \sigma_s + \sigma_c \cdot f\left(\frac{t - a_\sigma}{b_\sigma}\right)$$

$$\hat{\gamma}(t) = \gamma_s + \gamma_c \cdot f\left(\frac{t - a_\gamma}{b_\gamma}\right)$$

When applying non-stationary GEV distributions, it is often assumed that the only time-dependent parameters are location and scale, while the shape parameters is kept constant ([38, 57]). This approach leads us to a second type of model:

**Model 2a.** The GEV parameters location and scale are described using a logistic curve, using for each parameter a different initial value and amount of change. The parameters $a$ and $b$ are the same for location and scale. The shape parameter is kept constant over the whole time interval.

$$\hat{\mu}(t) = \mu_s + \mu_c \cdot f\left(\frac{t - a}{b}\right)$$

$$\hat{\sigma}(t) = \sigma_s + \sigma_c \cdot f\left(\frac{t - a}{b}\right)$$

$$\hat{\gamma}(t) = \gamma_{const}$$

**Model 2b.** This model is the same as Model 2a, but with individual parameters $a_\mu$, $b_\mu$, $a_\sigma$, $b_\sigma$ being used for location and scale of the GEV distribution.

$$\hat{\mu}(t) = \mu_s + \mu_c \cdot f\left(\frac{t - a_\mu}{b_\mu}\right)$$

$$\hat{\sigma}(t) = \sigma_s + \sigma_c \cdot f\left(\frac{t - a_\sigma}{b_\sigma}\right)$$

$$\hat{\gamma}(t) = \gamma_{const}$$

The logistic function, as used in the models above, has the limitation that the inflection point (the point of the strongest growth) is exactly in the middle of the curve, having always a value of $p_s + \frac{1}{2}p_c$. To allow for more flexibility, a generalized function that was proposed by Richards in [58] can be used. For $\beta > 0$, the Richards function is defined as

$$g_\beta(x) = \left(1 + (2^\beta - 1) \cdot \exp\left(-\log\left(\frac{0.95^{-\beta} - 1}{0.05^{-\beta} - 1}\right) \cdot x\right)\right)^{-\frac{1}{\beta}}. \tag{5}$$

We use it to describe the time-changing GEV parameters $p \in \{\mu, \sigma, \gamma\}$:

$$\hat{p}(t) = p_s + p_c \cdot g_\beta \left( \frac{t - a}{b} \right). \tag{6}$$

The interpretation of the parameters $p_s$ and $p_c$ remains unchanged. The parameter $a$ describes, as before, the time point at which the model attains the midpoint of the change (the value $p_s + \frac{1}{2}p_c$). In the previous models, this was also the point of the highest growth rate, while here, the inflection point depends on the value of the parameter $\beta$. For $\beta = 1$, the model reduces to the previous model ($g_1$ is equal to $f$), while the inflection occurs at a later time point than $a$ for $\beta > 1$ and at an earlier time point for $\beta < 1$. The parameter $b > 0$ controls the velocity of the change in such a way that the change from $p_s + \frac{1}{20}p_c$ to $p_s + \frac{19}{20}p_c$ (90% of the total amount of change) takes place in an interval of length $b$. Because of the asymmetry of the function $g_\beta$ for $\beta \neq 1$, this interval is no longer $\left[a - \frac{b}{2}, a + \frac{b}{2}\right]$, but shifted to the left for $\beta > 1$ and to the right for $\beta < 1$. In Fig 3, plots of the model function for different parameter values are shown.

Using the Richards function $g_\beta$ instead of $f$ in the previous models gives us four models that we denote by adding the letter R to the model name. Compared to the models using the function $f$, model 1aR and 2aR have one additional model parameter $\beta$, while model 1bR and 2bR feature additional model parameters for the non-constant GEV parameters $\beta_\mu$, $\beta_\sigma$ and (only Model 1bR) $\beta_\gamma$.

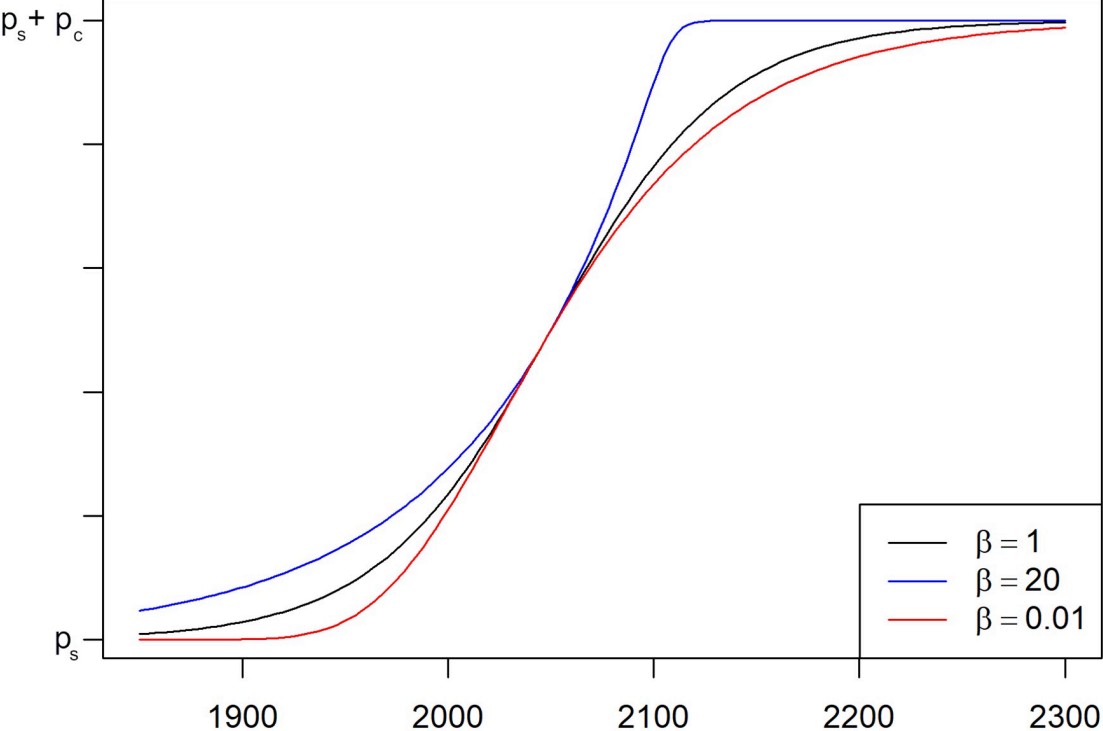

**Fig 3. Visualization of the parameter $\beta$ of the Richards function.** The plot shows Richards functions $g_\beta$ for different values of $\beta$. The Richards function for $\beta = 1$ is identical to the logistic function $f$. The point of the highest growth rate is shifted to the right for $\beta > 1$ and to the left for $\beta < 1$. For all lines shown, the other parameters used are $a = 2050$, $b = 100$.

## Model fitting and selection

Non-stationary GEV distributions can be fitted to data using Maximum Likelihood Estimators, see [59], Chapter 6.3 and [60]. In the numerical optimization, the fitting algorithm L-BGFS-B is used. For this purpose, the models are reparametrized to no longer use the parameters $p_c$ for $p \in \{\mu, \sigma, \gamma\}$ describing the magnitude of change, but parameters $p_e := p_s + p_c$ describing the values after the change instead. This makes it possible to ensure in an easy way that all values of $\sigma(t)$ are positive (using the condition $\sigma_e > 0$ instead of the equivalent $-\sigma_c < -\sigma_s$). To determine suitable starting values for the parameters $\mu_s$ and $\sigma_s$, a stationary GEV distribution is fitted to the first quarter of the data of the time series investigated, yielding estimates $\hat{\mu}$ and $\hat{\sigma}$, and starting values are selected randomly from the intervals $[\hat{\mu} - 5, \hat{\mu} + 5]$ and $[\max(0, \hat{\sigma} - 5), \hat{\sigma} + 5]$. The same is done for the parameters $\mu_e$ and $\sigma_e$ using the last quarter of the time series data. Since the estimation of the shape parameter is not very reliable for small samples, starting values for $\gamma_s$, $\gamma_e$ or $\gamma_{const}$ are not determined that way, but chosen randomly from the interval $[-1, 1]$. Random selection from an interval is also done for all other model parameters using suitable, large intervals to select values from. The stationary GEV distributions are fitted using the R package "EnvStats" ([61]). Part of our R-code is based on work by Takahito Mitsui in the context of [62]. The optimization algorithm is run several times with different starting values in order to find a global maximum of the likelihood function.

To choose the best model out of the different models presented here, we apply the Bayesian Information Criterion (BIC; [63]). To test the goodness-of-fit of the models, note that a GEV $(\mu, \sigma, \gamma)$-distributed random variable can be transformed to a GEV(1, 1, 1) distribution (a so-called unit Fréchet distribution) by applying the transformation

$$G_{\mu, \sigma, \gamma}(z) = \max\left(0, 1 - \gamma \cdot \left(\frac{z - \mu}{\sigma}\right)\right)^{-\frac{1}{\gamma}}. \tag{7}$$

By applying $G_{\hat{\mu}, \hat{\sigma}, \hat{\gamma}}(z)$ with the (time-dependent) estimated model parameters to the data, we obtain for each grid point a time series that is unit Fréchet distributed if the model assumptions are true. We test the hypothesis of the transformed data being unit Fréchet distributed using a one-sample Kolmogorov-Smirnov test ([64]).

## Proof of concept using simulated data

Before applying the statistical models to climate data, we first test how accurately model parameters can be estimated under ideal conditions. For each of the eight models presented above, we prescribe values for the model parameters, simulate data following the corresponding non-stationary GEV distribution, and fit the model to the data. We then compare the estimated model parameters with the real ones.

In addition to that, we test how susceptible the models that use the logistic function are to model misspecification. To this end, we simulate data from the models as above, but replacing the function $f$ from Eq (3) with the following three functions of a similar sigmoidal shape that

are known for example as activation functions for neural networks ([65, 66]):

$$g_1(x) \quad = \frac{1}{2} + \frac{1}{\pi}\arctan\left(\frac{\pi}{4}x\right) \tag{8}$$

$$g_2(x) \quad = \frac{1}{2} + \frac{x}{4\sqrt{1 + \frac{x^2}{4}}} \tag{9}$$

$$g_3(x) \quad = \frac{1}{2} + \frac{1}{2}\mathrm{erf}\left(\frac{x\sqrt{\pi}}{4}\right). \tag{10}$$

The simulated data follow a non-stationary GEV distribution with time-dependent distribution parameters $\mu_t$, $\sigma_t$, $\gamma_t$. We then calculate estimates $\hat{\mu}_t$, $\hat{\sigma}_t$, $\hat{\gamma}_t$ by fitting the original statistical model (using function $f$) to the data, and we calculate the time-integrated squared difference of given and estimated GEV parameters

$$\int_{t\in T} (p_t - \hat{p}_t)^2 \mathrm{d}t \tag{11}$$

for the three GEV parameters $p \in \{\mu, \sigma, \gamma\}$.

## Results

### Results of the simulation study

We simulated 1000 time series of length 150 for each logistic model using the parameters $\mu_s = 20$, $\mu_c = 10$, $\sigma_s = 2$, $\sigma_c = 1$. The parameters for the shape parameter were $\gamma_s = 0.1$ and $\gamma_c = 0.1$ (models with a varying shape parameter) or $\gamma_{const} = 0.1$ (models with a constant shape parameter). For the models describing a simultaneous change in all parameters we used the parameters $a = 2075$, $b = 30$, otherwise we used $a_\mu = 2050$, $b_\mu = 30$, $a_\sigma = 2075$, $b_\sigma = 30$ and, if applicable, $a_\gamma = 2100$ and $b_\gamma = 30$. The models based on the Richards function instead of the logistic function additionally had a parameter $\beta$ (or parameters $\beta_\mu$, $\beta_\sigma$, $\beta_\gamma$, respectively) equal to 5.

It turned out that for each parameter, the estimation quality is similar for all models in which the parameter occurs. In particular, the estimation is not more inaccurate for the more complex models with a higher number of parameters. For each parameter, boxplots of the estimates are shown in Fig 4. Since the estimates are similar for each model, only the boxplot for one model per parameter is shown. The boxplots indicate that the start and change values of the GEV parameters are in general well estimated, and the same is true for the parameter $a$ and $a_\mu$ if they exist in the model. The estimates for parameters $b$ and $b_\mu$ are in most cases close to the real value, but there are also some cases of a considerable misestimation (with a real parameter value of 30, the estimates take values of up to 120). The parameters describing a separate change in scale, $a_\sigma$ and $b_\sigma$, are estimated much worse than the other ones, estimates that are far away from the original value occur regularly. In addition, parameter $b_\sigma$ is in most cases underestimated, with the median of the estimates being far lower than the real value, while cases of a strong overestimation of this parameters also occur. The same can be said for the parameters $a_\gamma$ and $b_\gamma$, but their estimation accuracy is even lower.

The estimation of the additional $\beta$ parameters that appear in the models using the Richards function turned out to be very problematic for all models. The estimated values are usually far away from the real ones, and even the medians of the estimates are between 50 and 75 and not even close to the real parameter values of 5. A reliable estimation of the $\beta$ parameter of the

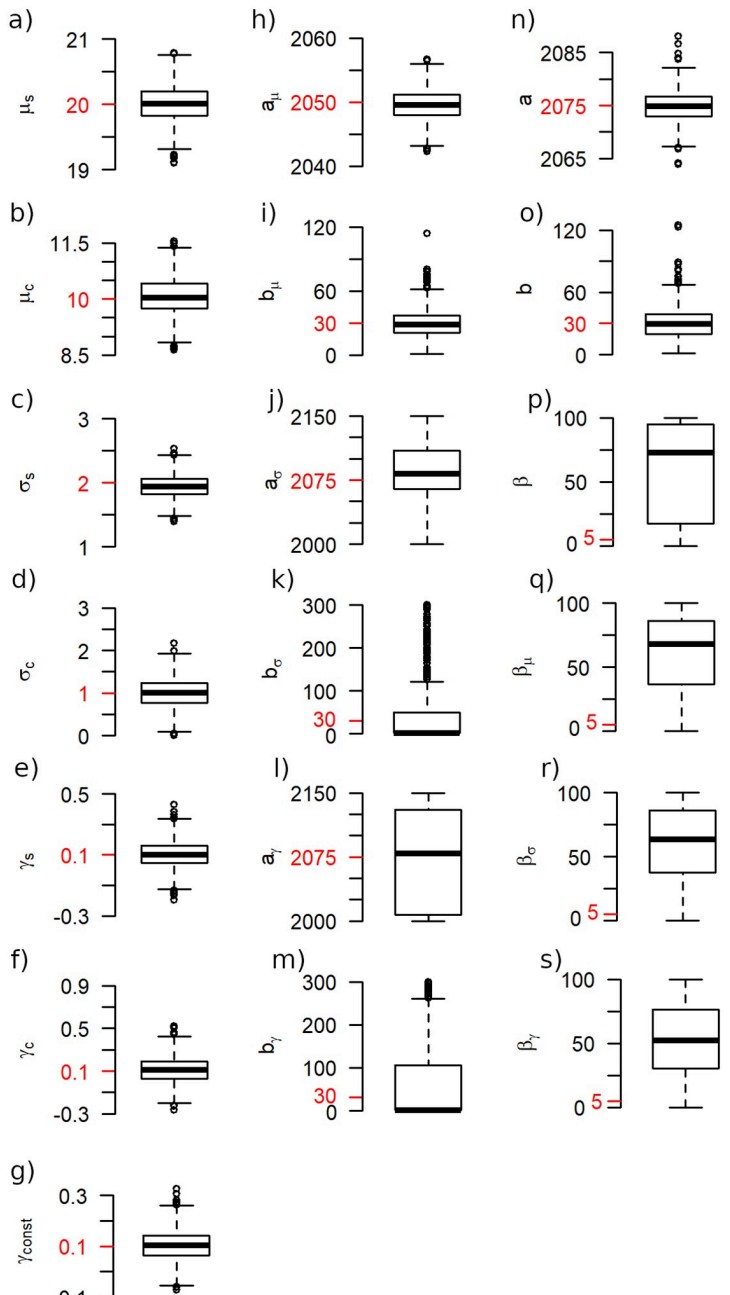

**Fig 4. The accuracy of the parameter estimation for simulated data.** The accuracy of the maximum likelihood estimators is investigated by applying the models to data that were generated following the respective model. For each parameter, a boxplot of the estimates is shown, with the real parameter value indicated in red. Since the results for each parameter are very similar across the models, only one boxplot is presented per parameter. The model depicted is 1a (a-f, n, o), 1b (h-m), 2a (g), 1aR (p) and 1bR (q-s).

Richards function seems to be impossible in general using the method we employed here. Because of that, the models using the Richards function will not be considered further and only the models using the logistic function will be applied to the data. It was considered also to exclude model 1b because of the high estimation inaccuracy of the parameters $a_\gamma$ and $b_\gamma$, but

**Table 1. The influence of model misestimation on the estimation accuracy.** The squared difference of constructed and estimated GEV parameters is shown for the GEV parameters location, scale and shape. The data were simulated using the sigmoidal function in the left-most column of the table while the model that was fitted to the data always uses the function $f(t)$. The model used is model 1a, results for the other models are similar. The errors are averaged over 5000 iterations.

| Function used | Location | Scale | Shape |
|:---:|:---:|:---:|:---:|
| $f(t)$ | 0.232 | 0.057 | 0.008 |
| $g_1(t)$ | 0.230 | 0.060 | 0.008 |
| $g_2(t)$ | 0.278 | 0.060 | 0.008 |
| $g_3(t)$ | 0.235 | 0.060 | 0.008 |

for the sake of completeness, the model was kept. In the next section it will be seen that this model is rarely favored by the BIC anyway.

The results of the simulation study investigating model misspecification due to other logistic functions than $f$ are shown in Table 1. Results are shown only for model 1a, but are similar for the other logistic models. For data that were created using one of the functions $g_i$, the errors are similar to those using function $f$. Therefore, model misspecification caused by the usage of different sigmoidal functions does not have a strong negative impact on the estimation accuracy and there is no need for using different functions than $f$ when applying the models to the data.

## Application to the data

As mentioned before, the only statistical models that are applied to the data are the four models based on the logistic function. We apply the models to four different climate simulations. In Fig 5, the best model according to the BIC is shown at each grid point for the four data sets.

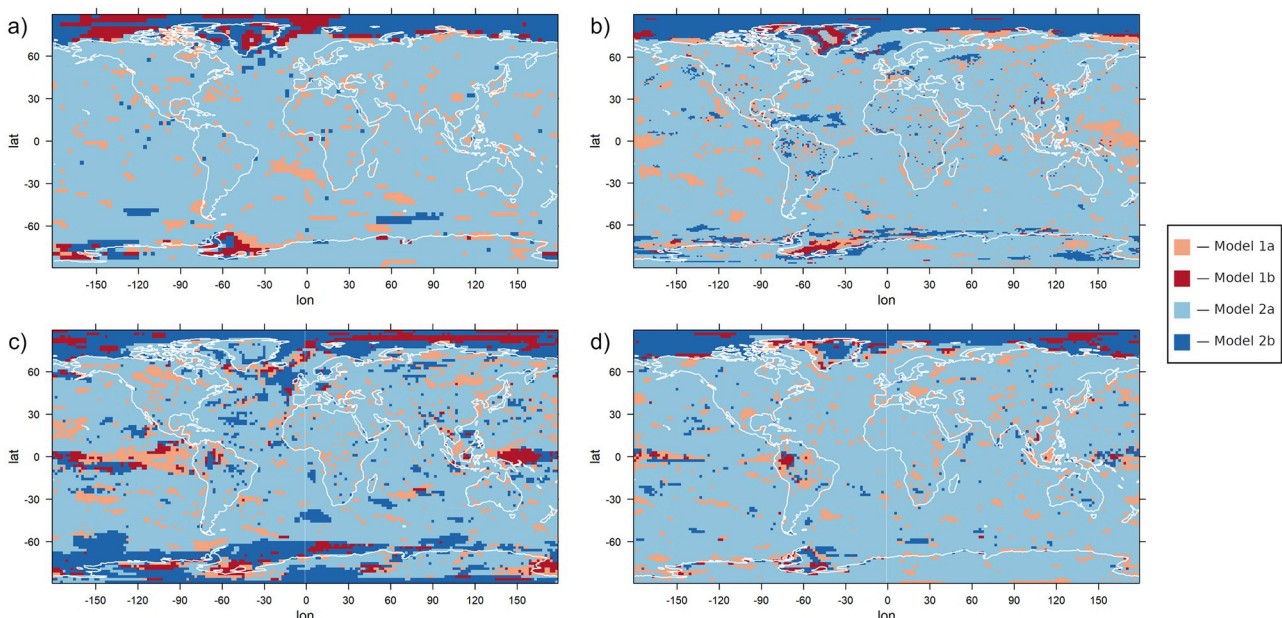

**Fig 5. The preferred model according to the Bayesian Information Criterion for each grid point.** The logistic models 1a, 1b, 2a and 2b are applied to the yearly maxima of daily temperature data and the BIC is used to determine the optimal one out of these for each grid point. Data set used: BCC (a), CCSM4 (b), CSIRO (c), MPI-ESM (d).

It can be noted that the statistical models with a constant shape parameter (model 2a and 2b) are often preferred over those with a varying shape parameter; one of these models is selected for at least 80% of the grid points for all data sets. There are many smaller regions in which a model with a varying shape parameter is preferred, a clear interconnection between those regions could not be identified. On the other hand, a pattern is visible regarding the question whether a model with simultaneous changes in location and scale (and, if applicable, shape) parameter is selected or not: Models with individual change parameters for the different GEV parameters are preferred almost exclusively in high-latitude regions. In particular, they are preferred throughout the whole region around the North Pole from ca. 80˚N onward for all four data sets, and for some data sets in a varying degree also in the high southern latitudes. In the other regions statistical models with a simultaneous change in the GEV parameters are predominant.

To investigate the magnitude of changes in extremes, in Fig 6, the difference of the 95% quantile of the fitted GEV distribution in the year 2300 and the quantile of the distribution in the year 1850 is depicted for each grid point and each data set. The statistical model used to calculate these values is the one that is preferred by the BIC at that grid point. While the magnitude of changes varies considerably depending on the data set, some general tendencies can be identified for all climate model outputs: The quantiles show in general an increasing trend, regions where the quantiles stay the same or decrease are an exception for all data sets. The quantile changes are higher over land than over the ocean, and in most data sets, particularly high changes can be detected in Europe, North America and parts of Siberia. Compared to the changes in other land regions of the world, Greenland shows an unusually small increase, in some models even partially a decrease.

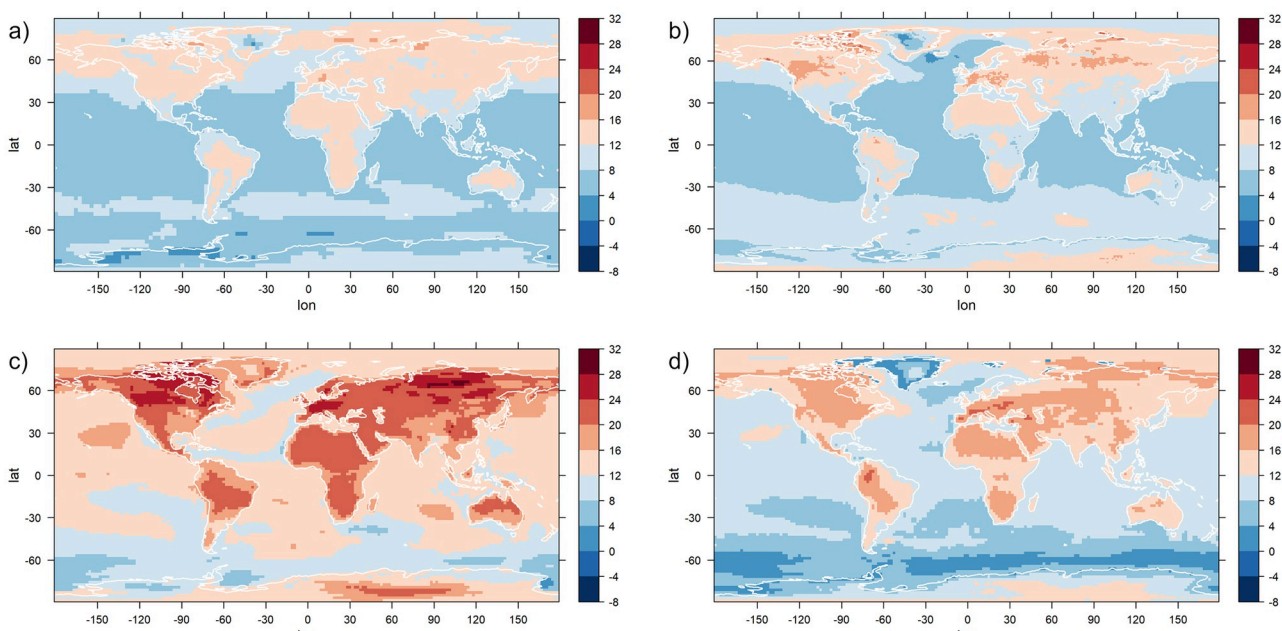

**Fig 6. Changes in the 95%-quantiles.** For each grid point, the change in the 95%-quantile of the fitted GEV distribution over the time interval from 1850 to 2300 is shown. The GEV distributions are estimated by fitting the logistic models to yearly maxima of daily temperature data. For each grid point, the statistical model that is preferred by the Bayesian Information Criterion is used. Units are ˚C. Data set used: BCC (a), CCSM4 (b), CSIRO (c), MPI-ESM (d).

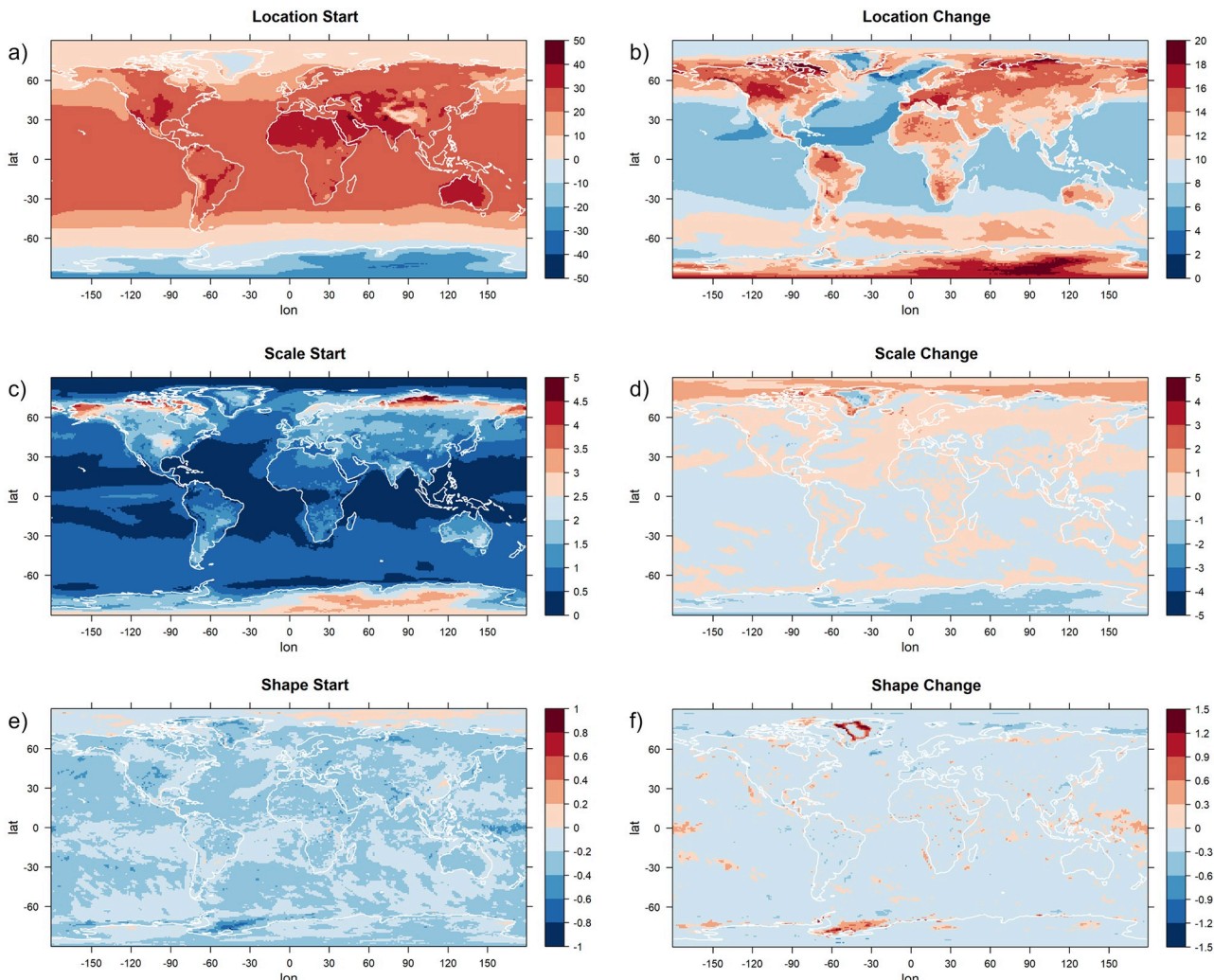

**Fig 7. The estimates for the starting values and the amounts of change of the GEV parameters.** The estimates for the parameters $\mu_s$ (a), $\sigma_s$ (c), $\gamma_s$ (e) and $\mu_c$ (b), $\sigma_c$ (d), $\gamma_c$ (f), describing starting value and total amount of change over time of the GEV parameters location, scale and shape. The models are applied to yearly maxima of daily temperature data of the climate model CCSM4. For each grid point, the estimates of the model that was preferred by the Bayesian Information Criterion are shown. Units are ˚C.

For the now following investigation of the individual parameters of the statistical models, plots are shown only for the data set CCSM4 (depicted in panel b of Figs 5 and 6), which has the highest resolution of the investigated data sets. The results for the other data sets are in general similar, significant deviations will be mentioned in text. Plots for those data sets can be found in the supplement (BCC: S1–S3 Figs. CSIRO: S4–S6 Figs. MPI-ESM: S7–S9 Figs).

All four statistical models we use share the parameters $\mu_s$, $\mu_c$, $\sigma_s$ and $\sigma_c$ describing the starting value and the magnitude of change of the location and scale parameters. Parameters $\gamma_s$ and $\gamma_c$ are estimated only for Models 1a and 2a, but for the other models we can define $\gamma_s$ as the constant estimate of the shape parameter and $\gamma_c$ as equal to zero. Using this definition, the values of the six parameters are shown in Fig 7. For each grid point, we show the estimates of the model that is preferred by the BIC at that grid point.

As expected, the starting value of the location parameter depends highly on the latitude and the climate zone of the grid point investigated. The starting values of the scale parameter show

a dependency on the continentality of the climate: the scale parameter is lowest over the oceans and highest in the very continental regions of Siberia, Alaska and northern Canada. It is also relatively high in Antarctica. The starting values of the shape parameter are quite homogeneous, attaining mostly slightly negative values that indicate that no strong positive extremes are present. The only exception to this are some regions in the Arctic Ocean, north of the regions with the high scale parameter discussed above. In these regions, the high values of the shape parameter together with low values of the scale parameter indicate a climate that is in general fairly homogeneous, but with occasionally strong outliers.

Investigating now the parameters describing the magnitude of change in the GEV parameters, we detect strong changes in the location parameter especially over land masses, with an increase of up to 20°C occurring in Europe and the central parts of North America. The highest changes, however, occur in the high-latitude regions that also feature a high initial shape parameter. Over the ocean, the changes in the location parameter are in general much smaller, especially in the Northern Hemisphere.

The scale parameter remains mostly the same in most regions, with a tendency to a slight increase. The most notable changes occur in the Arctic, where the scale parameter increases considerably, and in Antarctica, where it decreases. In most regions, the shape parameter is predicted to not undergo a change, as statistical models with a constant shape parameter are preferred by the BIC. Regions exhibiting a change in the shape parameter are in the Pacific near the equator and south of South America near Antarctica. A notable exception is a part of Greenland which shows not only a very strong increase in the shape parameter, but also unusually small increases in the location and the scale parameters.

Since a shift in the location parameter of a GEV distribution directly implies an equal shift in the quantiles, it is not surprising that the changes in 95% quantiles (Fig 6b) show a similar structure than the changes in the location parameter. The quantile changes are also affected by the changes in the scale parameter, therefore in Antarctica they are lower than the change in the location parameter would suggest (due to a decrease in the scale parameter), and in the Arctic they are higher (due to an increase in the scale parameter that is stronger than elsewhere).

We now turn our attention to the parameters describing at which time the changes take place. Models 1a and 2a have one parameter describing the time of change and one parameter describing its duration that are used for all three GEV parameters simultaneously. In Fig 8, these parameters are depicted. As before, for each grid point, the estimates of the statistical

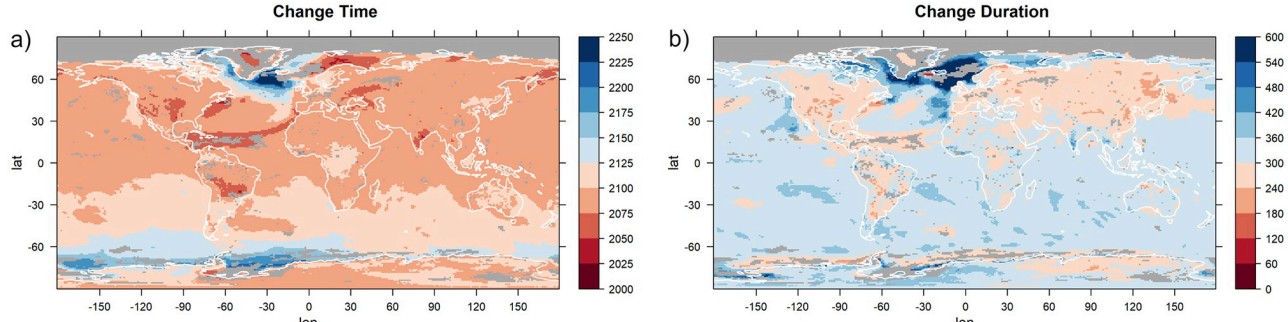

**Fig 8. The estimates for timing (a) and duration (b) of change, shown for statistical models with a simultaneous change in all GEV parameters.** The models are applied to yearly maxima of daily temperature data of the climate model CCSM4. For each grid point, the estimates of the model that was preferred by the Bayesian Information Criterion are shown. If the preferred model at a certain grid point does not feature parameters for simultaneous changes in the GEV parameters, the grid point is grayed out. Units are years.

model that was favored by the BIC are shown. If the selected model at a grid point is not one of model 1a or 2a, the grid point is grayed out. For most grid points, the time around which the change takes place is between 2075 and 2125 and the duration of the change is between 240 and 360 years. The most notable exception to this is the northern Atlantic Ocean, a region in which the duration of the changes tends to be much longer and highest change rate tends to occur much later. Changes that start unusually late occur also off the coast of Antarctica. Both regions are characterized by a ventilation of the deeper layers of the ocean providing an enhanced effective heat capacity dampening the warming signal (see e.g. [67]).

The other two statistical models, models 1b and 2b, have individual change parameters for the location and the scale (and, in the case of Model 1b, for the shape) parameter. These values are depicted in Fig 9 for the grid points at which one of those models is chosen. At all grid points, strong differences between the parameters corresponding to location and those

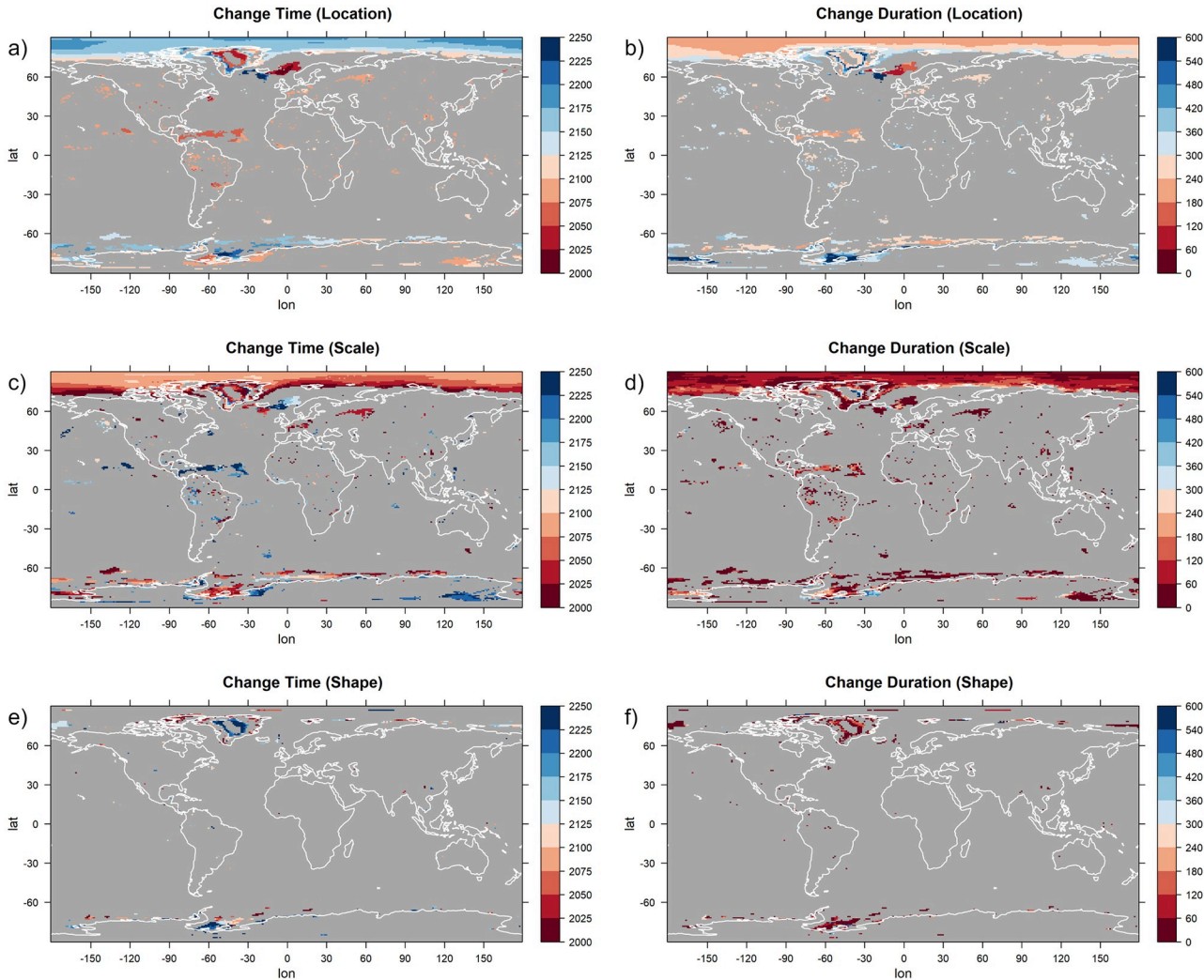

**Fig 9. The estimates for timing (a, c, e) and duration (b, d, f) of change for the location (a, b), the scale (c, d) and the shape (e, f) parameter, shown for statistical models with separate changes in the GEV parameters.** The models are applied to yearly maxima of daily temperature data of the climate model CCSM4. For each grid point, the estimates of the model that was preferred by the Bayesian Information Criterion are shown. If the preferred model at a certain grid point does not feature parameters for separated changes in the different GEV parameters, the grid point is grayed out. Units are years.

corresponding to scale can be seen, explaining why models that allow for individual changes in the different parameters perform better there. We focus on the largest contiguous region for which one of the models is selected, which is the area around the North Pole. In this region, changes in the scale parameter take place much earlier than those in the location parameter (2000–2100 compared to 2150–2200), and the scale parameter also changes considerably more rapidly than the location parameter (a duration of change of 0–120 years compared to 180–300 years).

To illustrate the four statistical models further, for each of them one grid point where the model is preferred by the BIC is selected. In Figs 10–13, the time series for those grid points are shown, together with the modeled time-dependent GEV parameters and the median and the upper and lower 95% quantiles of the modeled GEV distribution.

A first visual inspection indicates that the models seem to fit the data reasonably well. The most common model is Model 2a, showcased in Fig 12 for the grid point 0° N, 0° E. A clear logistic shape is visible in the time series of that grid point, which is reflected by a corresponding change over time of the location parameter. The scale parameter slightly decreases over time, while the shape parameter stays constant in this model. As already mentioned, Model 2a is preferred at most grid points, and the corresponding time series are usually similar to the one presented here.

While the shape parameter is constant in Model 2a, it undergoes a change over time in Model 1a, for which an example is shown in Fig 10 (grid point 0° N, 180° E in the Pacific Ocean). The shape parameter shows an increase over time here, while the scale parameter decreases at the same time. This indicates a shift to a climate with less variability in general, but more outliers than before. Model 1a is common in parts of the Pacific Ocean and it also appears in several small regions around the world.

Model 2b is predominant in the region around the North Pole, an example is depicted in Fig 13 for the grid point 85° N, 0° E. This model keeps the shape parameter constant and allows for sigmoidal changes in the location and scale parameters with different velocities and at different points in time. For the grid points near the North Pole, changes in the scale parameter are quicker and occur earlier than changes in the location parameter, as is also seen in Fig 13: In the first 200 years of the investigation period, the variability of the data is very low, and then it increases rather quickly in the years 2050 through 2100, while gradual changes in the location parameter follows later.

The fourth statistical model is model 1b, the most complex of the statistical models we use and the only one that allows for changes in all three GEV parameters at different speeds and points in time. There are only few regions where this model is preferred, one of them is a part of Greenland including the grid point 75° N, 35° W, which is analyzed in Fig 11. Besides the increase in the location parameter, we can detect here a strong decrease of the scale parameter that takes place mostly between 2000 and 2100. The shape parameter shows a pronounced increase after the year 2100. These model parameters indicate a complex behavior of the underlying time series that involves different kinds of changes at different points in time.

The goodness of fit of the statistical models is tested using a Kolmogorov-Smirnov test at significance level 5%, which is applied for each grid point to the results of the model that is preferred at that grid point by the BIC. There are only few grid point for which the hypothesis of the data following the modeled non-stationary GEV distribution is rejected, see Fig 14. It is important to keep in mind that the non-rejection of the hypothesis does not mean its confirmation, but still, this result is a promising indicator for the general applicability of the statistical models.

A detailed analysis of the different parameters of the statistical models was presented here for the earth system model CCSM4. For the other models, we briefly mention some key

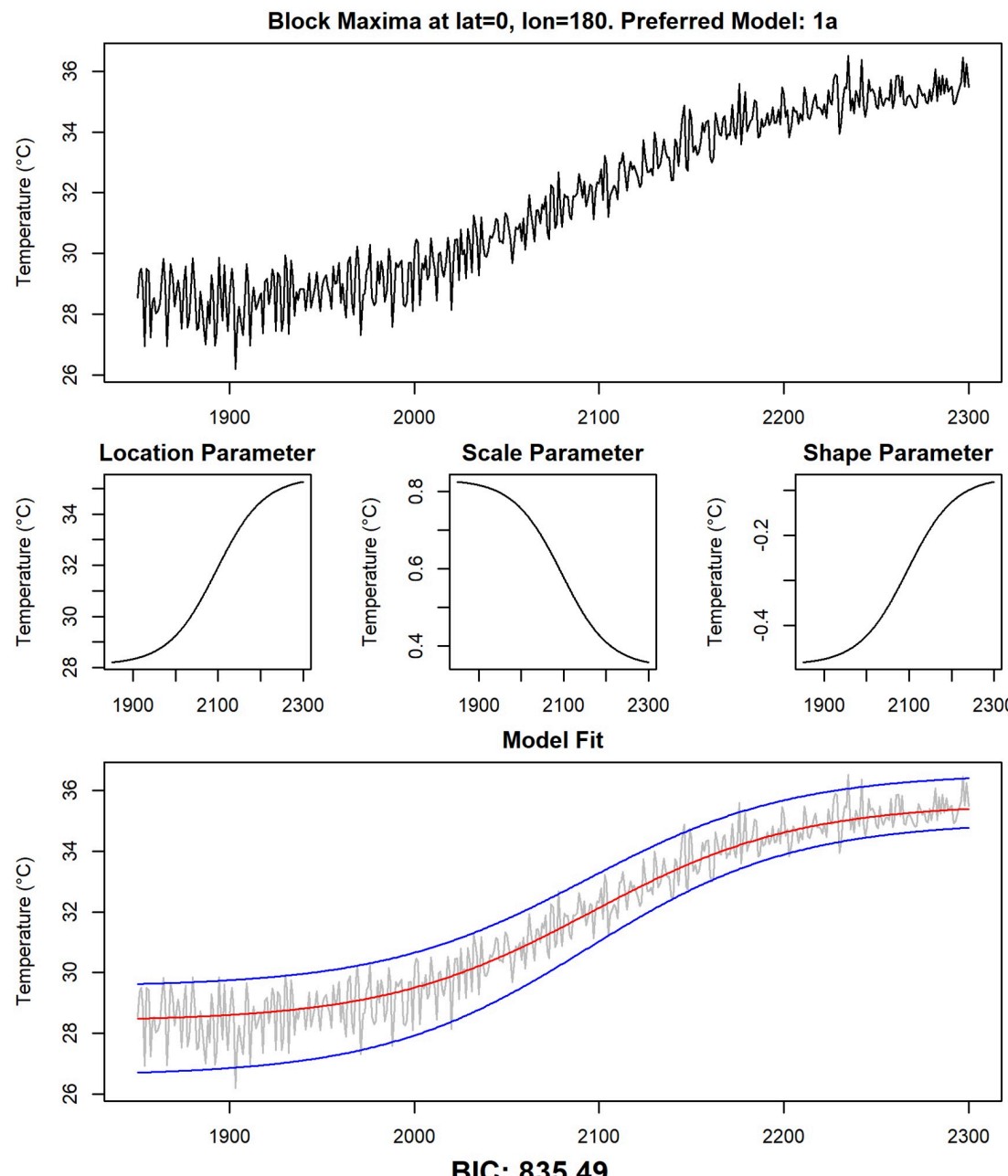

**Fig 10. Detailed examination of data and fitted models at grid point 0˚ N, 180˚E.** The yearly maxima of daily temperature data of the climate model CCSM4 at grid point 0˚ N, 180˚E, together with the non-stationary GEV parameter estimates of the preferred model at this grid point (model 1a) and the median of the estimated distribution (red line) as well as the 95% confidence interval (blue lines).

differences to the CCSM4 model. The estimated parameter values that are shown in Figs 7–9 for CCSM4 are shown for the other climate models in the supplement to this paper (S1–S9 Figs). The starting values of the three estimated GEV parameters are very similar for all four earth system models (compare Fig 7 and S1, S4, S7 Figs, panels a, c, e). CCSM4 tends to lead to higher values of the scale starting parameter in the high northern latitudes than the other models. This parameter also shows different values for Antarctica among the different earth system

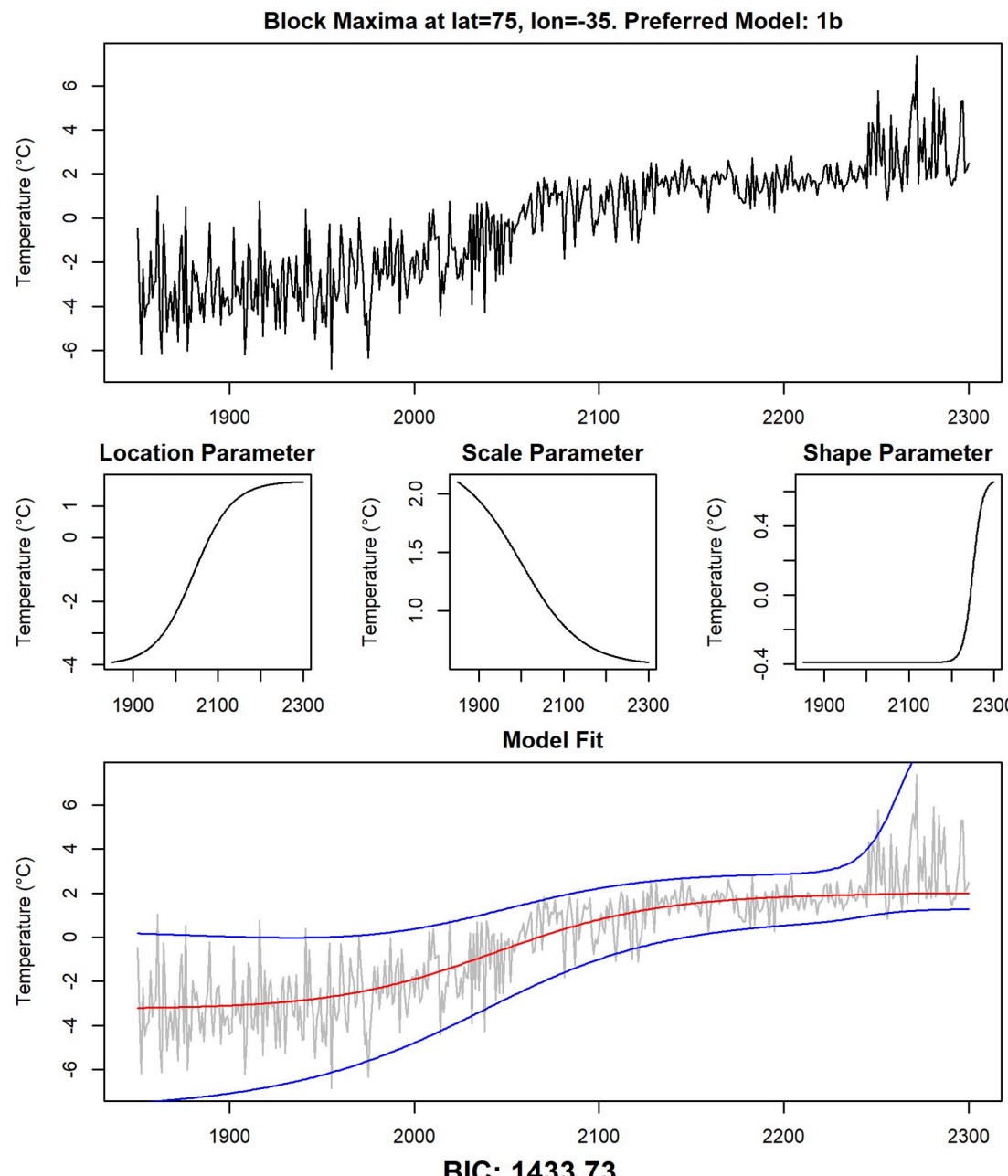

**Fig 11. Detailed examination of data and fitted models at grid point 75˚ N, 35˚ W.** The same analysis as in Fig 10 for grid point 75˚ N, 35˚ W (model 1b).

models. The changes in the location parameter (panel b of the figures) show a clear land-sea distinction in all four earth system models. In CSIRO, its values are considerably higher than in the other models, resulting also in the large difference in 95% quantiles compared to the other three models (Fig 6). This climate model also shows a region south of Africa near Antarctica with an unusually high scale change (panel d) and a high negative shape change (panel f) that is not identified in the other climate models. Other than that, all models agree that

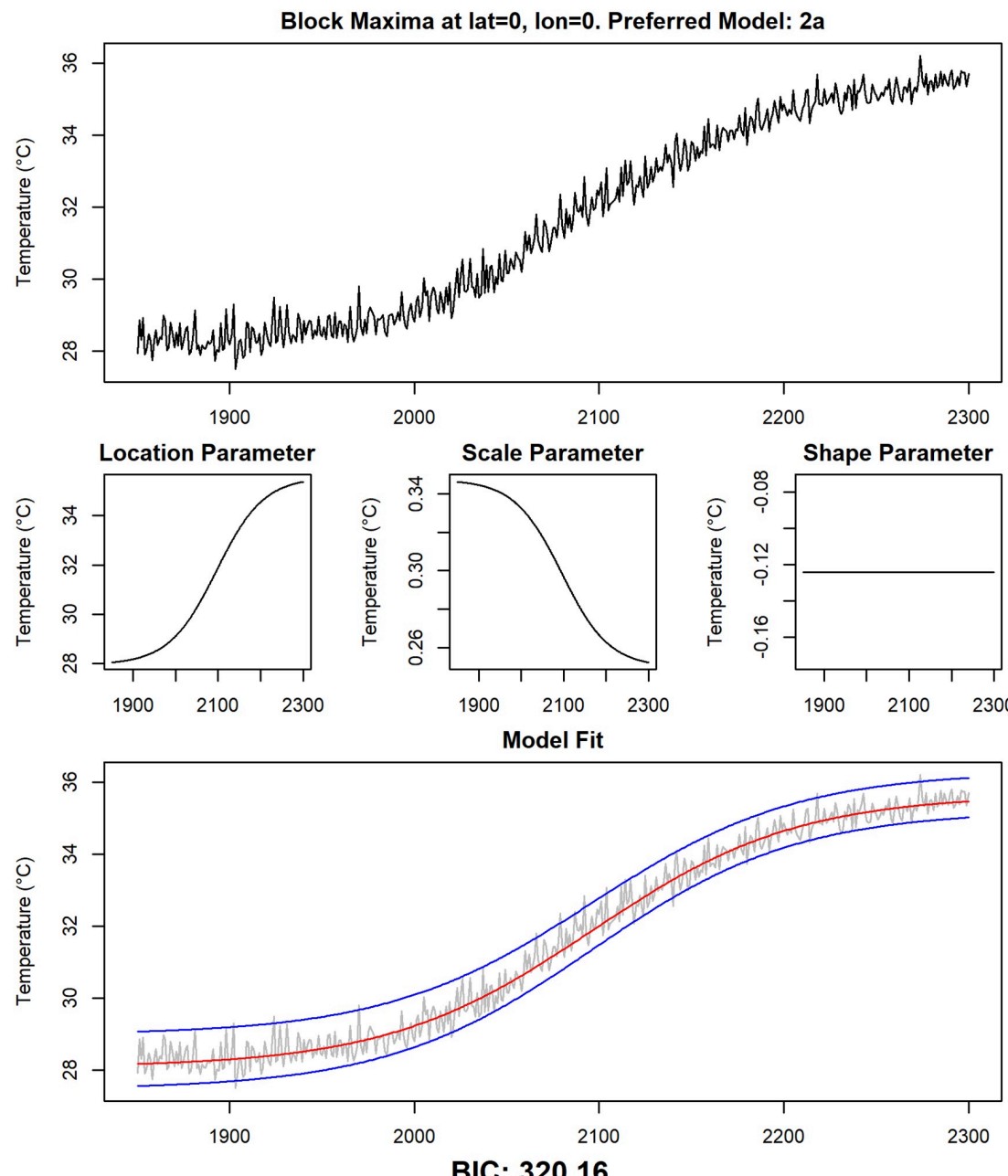

**Fig 12. Detailed examination of data and fitted models at grid point 0˚ N, 0˚ E.** The same analysis as in Fig 10 for grid point 0˚ N, 0˚ E (model 2a).

changes in scale are in general not high, with the exception of the high latitudes that show a strong increase in scale in the north and a decrease in scale in the south.

The timing of the simultaneous changes in all parameters also indicates a marked difference between CSIRO and the other climate models (compare Fig 8 and S2, S5, S8 Figs). In CSIRO, the time of the highest change rate (panel a of the figures) is in general approximately 50 years later than in the other models. Besides that, disagreements regarding the timing of changes exist also for the Indian and Pacific Ocean in the high southern latitudes, for which some

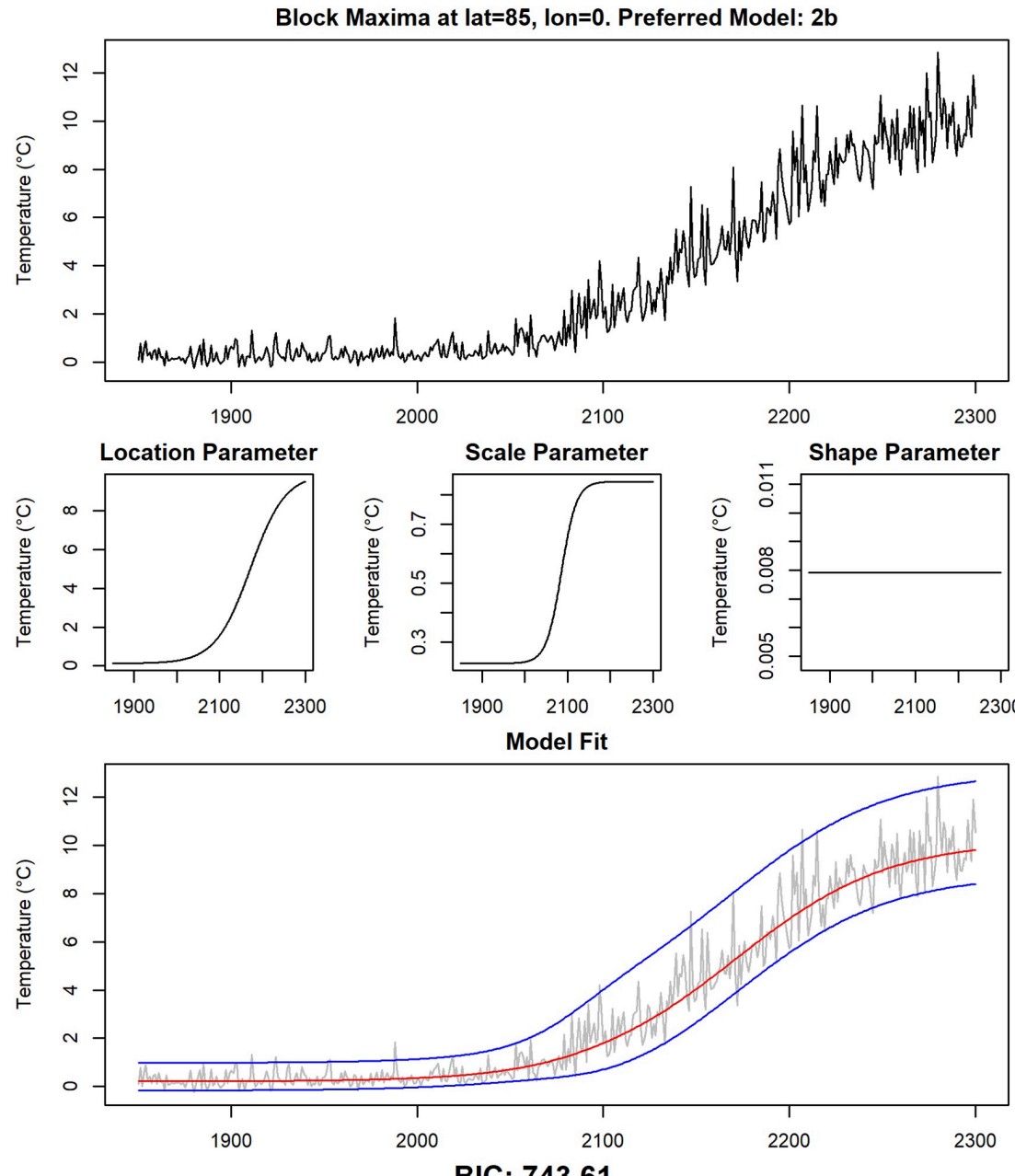

**Fig 13. Detailed examination of data and fitted models at selected grid point 85˚ N, 0˚ E.** The same analysis as in Fig 10 for grid point 85˚ N, 0˚ E (model 2b).

models predict a change that starts later and takes longer than in the CCSM4 model. The pro-longed changes in the North Atlantic Ocean that can be seen in panel b of Fig 8 for CCSM4 are not detected for the other earth system models (panel b of S2, S5, S8 Figs) and in general, it can be said that the four earth system models show large differences in the higher latitudes, both as to which statistical model is selected (see again Fig 5) and what parameter values are estimated. A common feature of all models is that near the North Pole, statistical models with separate changes in the location and the scale parameters are preferred and that the changes in

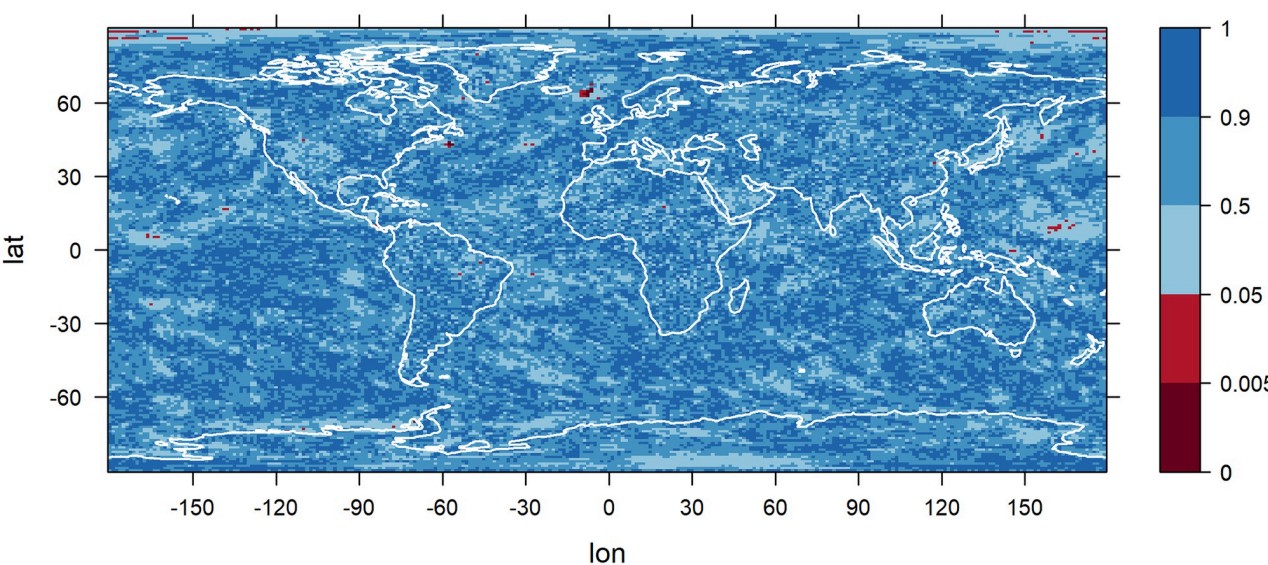

**Fig 14. P-values of the Kolmogorov-Smirnov test to investigate the goodness of fit.** A Kolmogorov-Smirnov test is applied for each grid point to the results of the statistical model that is preferred at that grid point by the BIC. The data used are yearly maxima of daily temperatures of the climate model CCSM4. Grid points at which the hypothesis of the data being GEV distributed with the modeled time-dependent GEV parameters is rejected at significance level 5% are colored in red.

scale precede the changes in location and also happen more quickly. The models disagree with regard to which regions use a constant and which ones use a variable shape parameter. For the CSIRO earth system model, separate changes in the location and the scale parameter are predicted in more regions than for the other three models, including large parts of Antarctica and the Pacific Ocean near the Equator (compare Fig 9 and S3, S6 and S9 Figs).

## Discussion

We present statistical models for extreme temperature that are applied to global climate data that span several hundred years and are influenced by climate change. While it is not a new approach to use non-stationary GEV distributions to investigate the development of climate extremes, most studies assume a dependency of the GEV distribution parameters on time that is either linear/polynomial ([39], [68–70]) or exponential ([71]). Consequently, the models are usually applied to data covering not more than 100 years. If the goal is to investigate changes in extremes on a longer time scale, the time frame is usually split up into several intervals of short length and stationary GEV distributions are fitted to each one. Then, their parameters are compared. This approach was used for global precipitation in [72], for precipitation and temperature in Australia in [73] and for summer temperature in the United States in [74]. In [75], annual maxima of daily temperature data from several CMIP6 models were investigated and stationary GEV distributions were fitted to the data at different time intervals. When stationary distributions are applied, it needs to be assumed that the changes in the investigated time intervals are not large. It it also more difficult to make statements regarding the temporal aspects of the changes. Non-stationary GEV distribution are advantageous in this regard, although it can be difficult to find suitable parametrizations for the parameters.

Our approach of combining logistic functions with GEV distributions to describe climate extremes has not been used before to our knowledge. Logistic functions have, however, been

used to describe historical $CO_2$ emissions in many countries ([76], [77]) and have also been applied to future projections of greenhouse gas concentrations ([78]). Climate change is closely connected to $CO_2$ concentrations, and the mean global temperature has been shown to be in an approximately linear relation to them ([79]). This further supports the idea of using logistic functions to describe extreme events under climate change as well.

The results in [75] that were obtained by fitting stationary GEV distributions to CMIP6 model results are mostly in line with the results of this study, even though we used a different greenhouse gas emission scenario (RCP8.5 vs. SSP370). In both studies, it is noted that the location parameter changes strongly over land and that this contributes to a large extent to the changes in extremes. A large increase in the shape parameter over the Arctic was detected in [75], and for the scale parameter, they identify a tendency for an increase over time in the tropics and a decrease over time in high-latitudes. Our study also shows an increase of scale in the low latitudes and a decrease of scale in Antarctica, but results for the Arctic are different. Our statistical models do not predict an increase in the shape parameter in the Arctic, but instead an incrase of the scale parameter while the shape parameter stays constant (Fig 7).

As discussed in the previous section, the Arctic region is unusual with regard to model selection: It is one of the few regions in which statistical models featuring non-simultaneous changes in location and scale parameter are preferred. In addition to that, the changes in the scale parameter are higher than in all other regions. Both results can be explained with the permanent presence of ice in the Arctic: the temperature of melting ice does not exceed 0˚C, therefore the annual maximum of daily temperature is close to 0˚C as long as ice is present all year long, resulting in a very low variability of the annual maxima. It is indeed shown ([80]) that in the RCP8.5 run of the CCSM4 earth system model, the Arctic becomes ice-free in the 2060s, which is also the period of time at which the variability of the time series starts to increase (Fig 13). After that, the value of the scale parameter is comparable to other land regions. This process also explains the complex behavior of the time series in Greenland in (Fig 11). Annual maxima are below 0˚C at the beginning of the investigation period. Due to increasing temperatures, ice begins to melt and the variability decreases, as the annual maxima are permanently close to 0˚C in the years 2100 through 2200. After that, ice has completely melted in summer and the variability increases again from 2200 onwards. This example also shows the limitations of the logistic models we present here: The increased variability in the years 2250 through 2300 is most likely better modeled by a high scale parameter than by a high shape parameter. This is also indicated by the results of fitting a stationary GEV distribution to the values of the time series in Fig 11 in the years 2250 through 2300. The fitted values are 1.44 for the scale parameter and 0.12 for the shape parameter. But it is not possible to model a change of the scale parameter going from a high value to a low value and then back to a high value again using a sigmoid function, so the BIC favors a model with a high shape parameter in later years instead.

It also needs to be emphasized that logistic functions are suitable for the modeling of future climate only under the condition of a cessation of greenhouse gas emissions in the future. For model data that are based on other scenarios, different functions have to be used, although logistic functions might also be useful for describing data that show a continuously rising trend in the extremes. In particular, the extraction and storage of atmospheric $CO_2$ in order to revert some consequences of climatic changes (and to prevent others) are more and more discussed. This is reflected by the SSP scenarios (replacing the RCP scenarios) used in the newer IPCC reports ([81]), of which some predict a reduction of the atmospheric $CO_2$ levels starting in the second half of the century. A possible extension of the logistic models for such a scenario

is based on the double logistic function

$$p_s + p_{c,1} \cdot f\left(2 \cdot \log(19) \cdot \frac{t - a_{p,1}}{b_{p,1}}\right) + p_{c,2} \cdot f\left(2 \cdot \log(19) \cdot \frac{t - a_{p,2}}{b_{p,2}}\right). \tag{12}$$

In this formula, two logistic function are combined, allowing for the description of a change from one state to another that is not completed, but instead reverted mid-way and that finally settles on an intermediate value. Such a model could also be useful with the RCP data sets used here to model the behavior in Greenland region for which sigmoidal functions are of limited suitability.

Besides that, the methodology presented in this work is not restricted to a specific application. The models or variations of them can also be applied to other data sets and other scientific questions regarding changes of extremes over time. Thus, the development of this methodology is a scientific contribution of its own right.

## Conclusions

In this work, we develop and apply statistical models for the development of temperature extremes over several centuries that allow us to investigate the magnitude and the timing of the changes in temperature extremes. In addition, the models differentiate between changes in the mean, the variability and the distributional shape of the estimated non-stationary GEV distributions. We summarize the conclusions of our work in the following main points:

1. A strong increase in the 95% quantiles of the annual temperature maxima could be detected in most regions of the world. In these regions, extremes will continue to rise and reach unprecedented strengths in the future. This is true especially over continents and corresponds to the well-known fact that global warming is stronger over land than over the oceans or in coastal regions ([16, 82]). However, we find a disagreement between the earth system models in terms of the total magnitude of the changes.

2. The development of extremes depends highly on the geographic region. Geographically varying developments can be detected not only with regard to the magnitude of changes, but also with regard to their timing, and to the extent to which the changes in extreme events are caused by changes in the location, the scale or the shape of the distribution of the annual maxima. For example, changes in the North Atlantic Ocean are slower than elsewhere.

3. Changes in location and scale of the distributions are predicted to take place simultaneously in most regions. Most earth system models agree that the highest rate of change is reached in the time between 2050 and 2100 over land and most parts of the oceans. In some high-latitude areas, changes over oceans start much later and last longer. The velocity of the change tends to be higher over land than over the oceans. Taking this together with conclusion 2, we can expect large and rather rapid changes in temperature over land masses over the course of about 100 years.

4. Non-simultaneous changes in the parameters are predicted is the region around the North Pole, in which an abrupt increase in variability is followed by a gradual increase of mean values. This is probably caused by the effects of the melting of sea ice. The earth system models disagree about the nature of the changes in Antarctica and in Greenland, which could hint to insufficient representations of polar processes in climate models such as feedbacks with the cryosphere (e.g. [83]). In addition, the statistical models presented here might not be suitable to describe the complex changes that are predicted for those regions.

## Outlook

Our focus is on the univariate analysis of temperature extremes: for each grid point, the time series of temperature data is investigated separately from all others. For a better understanding of climate extremes it is important to also investigate multivariate distributions. For example, climate extremes that take place simultaneously over a large region are especially problematic because of high damages for economies and possible difficulties in providing necessary medical or humanitarian aid. It would therefore be interesting to use spatio-temporal models to describe climate extremes. One possible class of those models are the max-stable models that are investigated for example in [84–86]. Since max-stable models require the data to be spatially stationary, i.e. that the joint distribution of two sites depend only on their geographical distance, they are usually only applied to small regions where such an assumption may be justified. Another approach to investigating spatial relations between climate extremes is the application of a clustering algorithm that identifies regions with similar extremal behavior ([87]) or the evaluation of atmospheric teleconnections ([13]). We plan to apply our methods presented here as a basis for the application of multivariate models in future work.

## Supporting information

**S1 Fig. As Fig 7, but for climate model BCC.**
(TIF)

**S2 Fig. As Fig 8, but for climate model BCC.**
(TIF)

**S3 Fig. As Fig 9, but for climate model BCC.**
(TIF)

**S4 Fig. As Fig 7, but for climate model CSIRO.**
(TIF)

**S5 Fig. As Fig 8, but for climate model CSIRO.**
(TIF)

**S6 Fig. As Fig 9, but for climate model CSIRO.**
(TIF)

**S7 Fig. As Fig 7, but for climate model MPI-ESM.**
(TIF)

**S8 Fig. As Fig 8, but for climate model MPI-ESM.**
(TIF)

**S9 Fig. As Fig 9, but for climate model MPI-ESM.**
(TIF)

## Acknowledgments

The authors wish to thank Manfred Mudelsee for constructive discussions and helpful suggestions. Thanks go to Takahito Mitsui for providing R-code and to Marcel Jacobse for giving advice for numerical optimization methods. We are also grateful to Lars Ackermann for providing support with the data acquisition and to the editoral team of PLOS ONE and two anonymous referees, whose constructive comments helped to improve the paper.

## Author Contributions

**Conceptualization:** Thorsten Dickhaus, Gerrit Lohmann.

**Investigation:** Justus Contzen.

**Methodology:** Thorsten Dickhaus, Gerrit Lohmann.

**Resources:** Gerrit Lohmann.

**Software:** Justus Contzen.

**Supervision:** Thorsten Dickhaus, Gerrit Lohmann.

**Writing – original draft:** Justus Contzen.

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
