## [Decision Letter · Decision Letter 0]

24 Oct 2022

PONE-D-22-20159Long-term temporal evolution of extreme temperature in a warming EarthPLOS ONE

Dear Dr. Contzen,

Thank you for submitting your manuscript to PLOS ONE. After careful consideration, we feel that it has merit but does not fully meet PLOS ONE’s publication criteria as it currently stands. Therefore, we invite you to submit a revised version of the manuscript that addresses the points raised during the review process.

We look forward to receiving your revised manuscript.

Kind regards,

Delei Li, Ph.D.

Academic Editor

PLOS ONE

Journal Requirements:

4. We note that Figures 5 to 9 in your submission contain [map/satellite] images which may be copyrighted. All PLOS content is published under the Creative Commons Attribution License (CC BY 4.0), which means that the manuscript, images, and Supporting Information files will be freely available online, and any third party is permitted to access, download, copy, distribute, and use these materials in any way, even commercially, with proper attribution. For these reasons, we cannot publish previously copyrighted maps or satellite images created using proprietary data, such as Google software (Google Maps, Street View, and Earth). For more information, see our copyright guidelines: http://journals.plos.org/plosone/s/licenses-and-copyright.

a. You may seek permission from the original copyright holder of Figures 5 to 9 to publish the content specifically under the CC BY 4.0 license.  

Reviewers' comments:

Reviewer's Responses to Questions

**Comments to the Author**

1. Is the manuscript technically sound, and do the data support the conclusions?

Reviewer #1: Partly

Reviewer #2: Yes

2. Has the statistical analysis been performed appropriately and rigorously? 

Reviewer #1: N/A

Reviewer #2: Yes

3. Have the authors made all data underlying the findings in their manuscript fully available?

Reviewer #1: Yes

Reviewer #2: Yes

4. Is the manuscript presented in an intelligible fashion and written in standard English?

Reviewer #1: Yes

Reviewer #2: Yes

5. Review Comments to the Author

Reviewer #1: In this manuscript. The authors just use a model, named as AWI-ESM, to discuss the extreme temperature change on global scale. In my opinion, the performace of global climate models in simulating the temperature changes show large uncertainties. Therefore, the uncertainty of the results in this manuscript could be large. The authors should use more outputs of global climate model to demonstrate the condifence of the results, and discuss the uncertainties of the results and methods. Due to this limitation, some minor comments and suggestions are not provided. The reviewer is looking forward to further improvement of this manuscript.

Reviewer #2: The study is focused on the statistical description of extreme monthly temperatures simulated iby one global climate model spanning the period 1850 through 2300. The authors apply a non-inflationary generalized extreme value distribution and prescribe a sigmoidal temporal evolution of the distribution parameters. The rationale is that changes in these parameters will be slow from 200 onwards, increase the rate of change later and settle to a new equilibrium once the concentrations of atmospheric greenhouse gases are stabilized.

The main conclusion is that land areas experience in the simulation more drastic changes of the extreme values than ocean areas with the exception of the Arctic Ocean. This is probably due to the disappearance, of sea ice cover in this region. In general, a non-stationary GEV is able to describe the temporal evolution of the distribution of extreme monthly temperatures reasonably well.

Recommendation: In my opinion, the manuscript is generally well written, the approach is sensible and complements other studies on non-stationary extremes. There are, though, a few points that came drew my attention, and that the authors may want to consider in a revised version.

Main points

1) the study is focused on monthly mean temperatures. Although this a choice made by the authors, it is somewhat strange, as usually extreme temperatures are defined from daily means or weekly means (e.g. heat waves). The reader may be therefore disappointed that this is not the case. This needs to be made explicit in the title. Also, the rationale for this choice is not clear, as the daily data are surely available and would give a more interesting information.

2) Although the text is generally well written, I would have wished sometimes a more accurate word choice. For instance, often the authors refer to models. The acquainted reader would be able to identify which types of models the authors are referring to (statistical, Earth System models..) but a more average reader would be slowed down when reading. I give some examples below.

3) The quality of the figures in the reviewer's copy is rather low - this may be a problem of the rendering by the submission system, but starting in Figure 6 and especially Figure 10 are of rather low quality. Figure 10 is barely readable.

Particular points:

3) I would open a new paragraph in line 15

4) line 96: From the year 2100 on, CO2 emissions are assumed to be zero.

I guess that the climate simulation does not include a carbon cycle model, at least this is not mentioned. Thus, the concentrations of greenhouse gases are prescribed and set to a constant value, Figure 4 also suggests this. This is different from prescribing the emissions to zero.

5) At several instances the text refers to 'long term' and 'long timescales'. Please, be more specific

on a long time-scale

6) Abstract and elsewhere 'Different models...' Please, specify (different statistical models)

7) ' In addition, our models differentiate between changes in mean, in variability and in distributional shape'

This can be a bit misleading, as the authors are referring to the mean, variability and shape of the GEV, but many readers will interpret this as changes in the mean, variability and shape of the original distribution of monthly temperatures. This is actually the most interesting point, whether for instance, changes in return values are caused by changes in mean temperature or by specifically changes in the tail of the distribution. In my interpretation this is not addressed in the study. Again., some readers may feel disappointed and this needs to be explicitly stated in the abstract.

8) line 14 ' Model simulations'. Please, specify Earth System models

9) 'Changes in the magnitude and the frequency of extreme events can be caused by

changes in the mean values, in the variability, in the heavy-tailedness or by a

combination of these factors'

see comment 7

10) ' the distribution of the maximum of independent and identically distributed

copies X1 , . . . , Xn of it'

Copies is not the right word, as their values are not identical

11) enumeration of equations ?

I think all equations should be numbered

12) line 385 : 2500 should read 2050 ?

6. PLOS authors have the option to publish the peer review history of their article (what does this mean?). If published, this will include your full peer review and any attached files.

Reviewer #1: No

Reviewer #2: No

---

## [Author Response · Author response to Decision Letter 0]

14 Dec 2022

Dear editors and reviewers,

thank you very much for the revision of our paper. Please see the attached PDF file "Response to Reviewers" for our answers to your comments.

---

## [Decision Letter · Decision Letter 1]

3 Jan 2023

Long-term temporal evolution of extreme temperature in a warming Earth

PONE-D-22-20159R1

Dear Dr. Contzen,

We’re pleased to inform you that your manuscript has been judged scientifically suitable for publication and will be formally accepted for publication once it meets all outstanding technical requirements.

Kind regards,

Delei Li, Ph.D.

Academic Editor

PLOS ONE

Additional Editor Comments (optional):

Reviewers' comments:

Reviewer's Responses to Questions

**Comments to the Author**

1. If the authors have adequately addressed your comments raised in a previous round of review and you feel that this manuscript is now acceptable for publication, you may indicate that here to bypass the “Comments to the Author” section, enter your conflict of interest statement in the “Confidential to Editor” section, and submit your "Accept" recommendation.

Reviewer #2: All comments have been addressed

2. Is the manuscript technically sound, and do the data support the conclusions?

Reviewer #2: Yes

3. Has the statistical analysis been performed appropriately and rigorously? 

Reviewer #2: Yes

4. Have the authors made all data underlying the findings in their manuscript fully available?

Reviewer #2: Yes

5. Is the manuscript presented in an intelligible fashion and written in standard English?

Reviewer #2: Yes

6. Review Comments to the Author

Reviewer #2: I thank the authors for considering my comments when revising the manuscript. I am now happy to recommend its publication

7. PLOS authors have the option to publish the peer review history of their article (what does this mean?). If published, this will include your full peer review and any attached files.

Reviewer #2: No

---

## [Editor Report · Acceptance letter]

6 Jan 2023

PONE-D-22-20159R1 

Long-term temporal evolution of extreme temperature in a warming Earth 

Dear Dr. Contzen:

I'm pleased to inform you that your manuscript has been deemed suitable for publication in PLOS ONE. Congratulations! Your manuscript is now with our production department. 

Kind regards, 

on behalf of

Dr. Delei Li 

Academic Editor

PLOS ONE